# Low-Temperature Enhancement-Mode Amorphous Oxide Thin-Film Transistors in Solution Process Using a Low-Pressure Annealing

**DOI:** 10.3390/nano13152231

**Published:** 2023-08-01

**Authors:** Won Park, Jun-Hyeong Park, Jun-Su Eun, Jinuk Lee, Jeong-Hyeon Na, Sin-Hyung Lee, Jaewon Jang, In Man Kang, Do-Kyung Kim, Jin-Hyuk Bae

**Affiliations:** School of Electronic and Electrical Engineering, Kyungpook National University, Daegu 41566, Republic of Korea; qkrdnjs0320@knu.ac.kr (W.P.);

**Keywords:** metal-oxide semiconductor, thin-film transistors, solution process, low processing temperature, enhancement mode, low-pressure annealing

## Abstract

The interest in low processing temperature for printable transistors is rapidly increasing with the introduction of a new form factor in electronics and the growing importance of high throughput. This paper reports the fabrication of low-temperature-processable enhancement-mode amorphous oxide thin-film transistors (TFTs) using the solution process. A facile low-pressure annealing (LPA) method is proposed for the activation of indium oxide (InO_x_) semiconductors at a significantly low processing temperature of 200 °C. Thermal annealing at a pressure of about ~10 Torr induces effective condensation in InO_x_ even at a low temperature. As a result, the fabricated LPA InO_x_ TFTs not only functioned in enhancement mode but also exhibited outstanding switching characteristics with a high on/off current ratio of 4.91 × 10^9^. Furthermore, the LPA InO_x_ TFTs exhibit stable operation under bias stress compared to the control device due to the low concentration of hydroxyl defects.

## 1. Introduction

Oxide thin-film transistors (TFTs) are regarded as one of the most important building blocks for applications such as displays, optoelectronics, and back-end-of-line transistors [1,2,3,4]. Recently, the solution process for oxide TFTs has attracted a lot of attention because it enables large-area manufacturing and high throughput [5,6,7]. However, compared to conventional sputtering-based oxide TFTs, solution-processed oxide TFTs not only exhibit relatively low field effect mobility (μ_FE_) and electrical stability but also require a high processing temperature (>400 °C) [8]. In particular, high processing temperature leads to limitations in the application of flexible polymer substrates, making it difficult to utilize in next-generation electronics with new form factors [6]. Unfortunately, the solution-processed oxide TFTs fabricated at a low temperature of 200–300 °C exhibit extremely low μ_FE_ and electrical stabilities.

Binary oxides, such as tin oxide and indium oxide (InO_x_), were studied to address these issues [9,10]. Although these semiconducting materials can be used to obtain high mobility, a high electron concentration of these materials cause the negative threshold voltage (V_T_), thereby resulting in depletion-mode TFTs. However, the positive V_T_ is suitable for the pixel and gate driver circuits because enhancement-mode characteristic simplifies the circuit design and reduces power consumption [11,12]. Moreover, binary oxides typically have poly- or nanocrystalline structures [9,13,14]. Fabricating enhancement-mode oxide TFTs through low processing temperatures in the solution process is urgently required.

This work seeks to address the aforementioned issues by reporting the fabrication of enhancement-mode amorphous oxide TFTs at a low processing temperature of 200 °C using a solution process. A low-pressure annealing (LPA) method is introduced to achieve the activation and densification of solution-processed InO_x_ semiconductors at a significantly lower processing temperature of 200 °C. Thermal annealing at a low pressure of about ~10 Torr helps in diffusing out hydrogen even at low temperatures, resulting in an effective condensation reaction. Considering that the vacuum of thermal evaporation used for organic light-emitting diode deposition has a pressure of ~10^−6^ Torr, an efficient semiconductor activation can be achieved with a relatively weak vacuum level. As a result, the LPA InO_x_ TFTs not only functioned in the enhancement mode but also exhibited excellent switching characteristics with a high on/off current ratio of 4.9 × 10^9^. Furthermore, compared with the control devices, the LPA InO_x_ TFTs exhibited stable operation under positive and negative gate bias stress compared with the control devices.

## 2. Materials and Methods

A 0.1 M InO_x_ solution was prepared by dissolving In(NO_3_)_3_⋅xH_2_O (Sigma-Aldrich, St. Louis, MO, USA) in 2-methoxyethanol to fabricate InO_x_ semiconductor thin films. The solution was stirred at 50 °C for 3 h to obtain a clear and homogeneous solution. A 100 nm thick Si/SiO_2_ wafer substrate was prepared for the gate electrode and dielectric through cleaning with sonication in acetone, isopropyl alcohol, and deionized (DI water) for 10 min each. The residual moisture present on the wafers was fully removed by N_2_ and annealed at 350 °C. Figure 1 shows a schematic of the fabrication process of oxide TFTs. For the patterning of InO_x_ semiconductors on SiO_2_ dielectric, the water etchant-based photopatterning method was used, which was reported previously [15]. First, a 0.1 M InO_x_ solution was spin-coated on the SiO_2_ dielectric at 3000 rpm for 20 s to create a film thickness of 7 nm. The precursor solution-deposited substrate was soft-baked at 100 °C for 1 min. Afterward, to avoid ozone generation, the sample covered with a fine metal mask was exposed to ultraviolet (UV) radiation (25 mW cm^−2^) from a low-pressure mercury lamp with two main wavelengths of 253.7 nm (90%) and 184.9 nm (10%) in N_2_ ambient for 150 s. In this stage, the photochemical cleavage of alkoxy groups and the decomposition of nitrate ligands were achieved in InO_x_ [15,16]. Consecutively, the substrate was developed in DI water etchant for 1 min. Etching through water supplies additional water molecules to InO_x_ and promotes the condensation reaction in the subsequent annealing process [15,17]. Pre-annealing was conducted at 100 °C in atmospheric conditions for 10 min. To fabricate the control device, which was annealed in an NPA (normal pressure annealing) environment, it was post-annealed in an oven at 200 °C for 2 h without vacuum (760 Torr). To fabricate the low-pressure annealed InO_x_ based TFTs, InO_x_ was post-annealed in an oven at a low-pressure environment of 10 Torr for 2 h at 200 °C. The InO_x_ films annealed under normal- and low-pressure are called NPA and LPA InO_x_. Finally, the 50 nm-thick Al was deposited using thermal evaporation for source and drain electrodes. The channel length (L) and width (W) were 100 and 1000 μm, respectively. X-ray photoelectron spectroscopy (XPS) (NEXSA, Thermo Fisher Scientific, Waltham, MA, USA) was used to analyze the chemical composition of the NPA and LPA InO_x_ thin films. Grazing incidence X-ray diffraction (GIXRD) (DMAX-2500-PC, Rigaku, Tokyo, Japan) was performed to characterize the crystal structure of InO_x_. X-ray reflectance (XRR) (ATX-G, Rigaku) measurements were performed to investigate the film density of NPA and LPA InO_x_. The dielectric properties of NPA InO_x_ and LPA InO_x_ were analyzed using a probe station (Model 4000, MS Tech, Arlington, VA, USA) equipped with a precision LCR meter (Keysight, Santa Rosa, CA, USA) and a semiconductor parameter analyzer (Keithley 2636 B, Tektronix Inc., Beaverton, OR, USA).

## 3. Results and Discussion

### 3.1. Effect of LPA on Atomic Bonding State of InO_x_

XPS analysis was conducted to illustrate the impact of LPA on the atomic composition in the InO_x_ semiconductor. Figure 2a,b presents the XPS C 1s and In 3d spectra of NPA and LPA InO_x_ thin films, respectively. Both NPA and LPA InO_x_ thin films show clear peaks at 284.6 and 445.5 eV for C 1s and In 3d, respectively, and there is negligible difference in peak intensity. In addition, NPA and LPA InO_x_ thin films show a similar atomic ratio of C, N, O, and In, as shown in Figure 2c. This result indicates that the LPA process does not induce the diffusing out of byproducts, such as nitrogen or carbon. Figure 2d,e shows the deconvoluted XPS O 1s spectra of NPA InO_x_ and LPA InO_x_, respectively. Three main peaks were observed at the ~529.6, ~530.5, and ~531.5 eV positions, corresponding to metal–oxygen (M–O), oxygen atoms near oxygen vacancies and in M–OC bonds (V_o_ + C–O), and metal hydroxide (M–OH), respectively [16,18,19,20]. Compared to NPA InO_x_, LPA InO_x_ showed a relatively increased ratio of M–O bonding to V_o_ + C–O, and in particular, M–OH bonding was significantly reduced, as shown in Figure 2d,e. NPA InO_x_ and LPA InO_x_ have M–OH bonding of 33% and 12.6%, respectively (Figure 2f). In the process of etching with pure water, a large amount of M–OH was formed in InO_x_ due to the additional supply of water molecules by the water etchant. The annealing temperature is not high enough for the condensation reaction. As a result, M–OH still remains in InO_x_ [21]. On the other hand, the low M–OH ratio in LPA InO_x_ results from the diffusing out of hydrogen via thermal annealing in a vacuum environment [22]. Although annealing temperature is 200 °C the same as NPA InO_x_, the vacuum environment assists in the effective removal of hydrogen, thereby leading to an increase in the M–O bond. The M–O increases from 38.0% to 45.2% through LPA. In general, in oxide semiconductors, hydroxyl acts as an acceptor-like trap and hinders electron conduction, while the M–O network provides an electron transport path through the overlap of metal s orbitals [23]. Therefore, it is necessary to minimize the M–OH and enhance the M–O network in the oxide semiconductor for high-performance electronic devices. Meanwhile, vacuum induces not only the diffusing out of hydrogen but also the desorption of oxygen from the surface [24]. This desorption of oxygen consequently promotes the formation of V_o_ + C–O and improves the conductivity of the semiconductor. The introduction of LPA led to an increase in V_o_ + C–O from 30.0% to 42.2%.

### 3.2. Physical Properties of InO_x_ Semiconductor Thin Films

XRR analysis was performed to investigate the effect of LPA on the density of InO_x_ semiconductor films. Figure 3a shows the XRR spectra of NPA and LPA InO_x_. The mass density of a thin film is related to the critical angle (*θ_c_*). Equation (1) shows that there is a proportional relationship between the density and *θ_c_* [25]:(1)$$\theta_{c}^{2}=(\frac{e^{2}\lambda^{2}}{\pi mc^{2}})\cdot (\frac{N_{A}Z}{A})D$$
where *N_A_* is Avogadro’s number, λ is the X-ray wavelength, Z is the average number of electrons per atom, A is the average atomic mass, and D is the mass density of the film. As shown in Figure 3a, the critical angle of LPA InO_x_ is positively shifted. The extracted film densities of NPA InO_x_ and LPA InO_x_ are 5.19 and 5.45, respectively, indicating that the LPA treatment increases the film density. In other words, the LPA treatment induces dense M–O bonds by diffusing out of hydrogen ions and suppresses physical defects, such as pinholes and pores [26,27]. This result agrees with the XPS analysis.

Figure 3b shows the GIXRD spectra for NPA InO_x_ and LPA InO_x_ thin films. No clear peaks were observed for both NPA InO_x_ and LPA InO_x_, indicating that the thin films have an amorphous phase [28]. Binary oxide semiconductors, such as InO_x_, typically have poly- or nanocrystalline structures, even at low processing temperatures [9]. Furthermore, since the poly- or nanocrystalline structure of semiconductors has grain boundaries, the yield and uniformity of devices in mass production are decreased. Therefore, the amorphous phase could be preferred in the industry. The amorphous phase of NPA InO_x_ and LPA InO_x_ might be attributed to the fact that additional water molecules are supplied in the initial stage of thin film formation through the water etching process to hinder InO_x_ crystallization [15,29].

The proposed chemical mechanism of LPA effect on InO_x_ is illustrated in Figure 4. NPA InO_x_ contains hydrogen-related defects as well as the M-O network due to insufficient thermal energy (Figure 4a). These OH-related defects not only lead to an increase in the electron concentration in the semiconductor but also act as acceptor-like traps [11,30]. In addition, the high ratio of M-OH weak bonds and the low mass density is induced due to insufficient condensation and densification. Such an imperfect atomic structure of NPA InO_x_ can be understood through the results of XPS and XRR, which were discussed above. On the other hand, in LPA InOx, film activation based on the condensation reaction is effectively achieved due to the diffusing out of hydrogen in the weakly bonded O–H (Figure 4b). As a result, M–O bonding increases and defects decrease, thereby forming a favorable semiconducting film. The following Equation (2) represents the condensation reaction:M–OH + M–OH → M–O–M + H_2_O(2)

### 3.3. Electrical Characteristics of LPA InO_x_ TFTs

The transfer and output characteristics of the device were investigated to examine the effect of LPA on the electrical properties of InO_x_ TFTs. Figure 5a,b shows the transfer characteristics of NPA and LPA InO_x_ TFTs, respectively. To obtain the transfer curves, the gate voltage was increased from −10 to 30 V, while the drain voltage was maintained at 30 V. Surprisingly, the µ_FE_ of the LPA InO_x_ TFTs showed a high value of 0.81 cm^2^ V^−1^ s^−1^ despite the low processing temperature and amorphous phase. This value is about six times higher than the µ_FE_ of NPA InO_x_ TFT, 0.13 cm^2^ V^−1^ s^−1^. Note that there is a trade-off relationship between annealing temperature and mobility, and V_th_ and mobility in oxide semiconductor system due to the charge transport mechanism [15,31]. Thus, the µ_FE_ of oxide TFTs in this study could be dramatically increased by increasing the annealing temperature or semiconductor thickness, although this study is focused on the low processing temperature and enhancement-mode operation. Furthermore, the introduction of LPA leads to an increase in the on/off current ratio from 6.83 × 10^7^ to 4.91 × 10^9^. The subthreshold slope (SS), which is related to the interface trap density, also greatly decreased from 0.71 V dec^−1^ to 0.28 V dec^−1^. Considering the above chemical and physical analysis results, the increase in M–O bonds acting as a charge transport path and the decrease in M–OH bonds serving as trap sites through the LPA treatment are the dominant factors that cause the changes in the electrical properties. Figure 5c,d shows the output characteristics of NPA and LPA InO_x_ TFTs, respectively. The output curves were measured by applying the gate voltage from −10 to 30 V in 10 V steps. Drain voltage was swept from 0 to 30 V. LPA InO_x_ TFT showed significantly higher drain current than NPA InO_x_ TFT under the same voltage conditions, and both devices showed clear linear and saturation regions. The electrical parameters of LPA oxide TFTs and recently reported oxide TFTs are summarized in Table 1.

### 3.4. Bias Stress-Induced Instability of LPA InO_x_ TFTs

Since the bias stress-induced instability directly affects the lifetime of the device, ensuring bias stability is one of the most important factors in determining its commercialization. The effects of LPA on the stability of TFTs under positive bias stress (PBS) and negative bias stress (NBS) are investigated. Figure 6a,b shows the PBS time-dependence of transfer curves of NPA and LPA InO_x_ TFTs, respectively. Gate and drain voltages of 10 V and 0 V, respectively, were applied to the PBS test. The ΔV_th_ of NPA and LPA InO_x_ TFTs were 6.20 and 4.40 V, respectively (Figure 6c). LPA InO_x_ TFTs exhibited superior PBS stability compared to NPA InO_x_ TFTs. In both devices, SS does not change as the stress time increases, and only a positive V_th_ shift was confirmed. This suggests that the degradation caused by PBS is due to electron trapping at gate dielectricthe presence of InO_x_ semiconductor interfaces and electron injection into gate dielectric rather than defect creation [38,39]. Since the thermally grown high-quality SiO_2_ was used for the gate dielectric of NPA and LPA InO_x_ TFTs, the improvement in ΔV_th_ through LPA implies the reduction in electron trapping at the interface. Recently, it has been revealed that excess hydrogen peroxide could be the origin of PBS-induced V_th_ degradation [4]. However, since solution-processed InO_x_ is characterized by a relatively oxygen-poor configuration, unlike IGZO sputtered at high oxygen flow rates, oxygen-related defects cannot be considered the origin of PBS-induced ΔV_TH_. Therefore, it is reasonable that the reduction in hydrogen-related defects through LPA leads to the improvement of PBS in oxide TFTs based on the XRR and XPS results. Figure 6c,d shows the NBS time-dependent transfer curves of NPA and LPA InO_x_ TFTs, respectively. The gate and drain voltages of −10 V and 0 V, respectively, were utilized for the NBS test. After 3000 s of NBS, the ΔV_th_ of NPA and LPA InO_x_ TFTs were −1.8 and −0.91 V, respectively, and LPA InO_x_ TFTs showed superior NBS stability compared to NPA InO_x_ TFTs (Figure 6f). Hole trapping, oxygen vacancy, and excess hydrogen peroxide models have been considered mechanisms of V_th_ deterioration via NBS [4,38,39]. It is known that SS deterioration generally occurs based on oxygen vacancy or peroxide model under a strong negative gate bias or illumination stress [4,38,40]. Since the negative shift of V_th_ is observed without changing SS in this study, the shift of V_th_ in NPA and LPA InO_x_ TFTs might be attributed to hole trapping and injection rather than the ionization of oxygen vacancy. Therefore, the superior NBS stability of LPA InO_x_ TFTs originates from the reduction in organic chemical- and hydrogen-related defects, which could act as hole-trapping states [26,41]. Although the PBS and NBS were analyzed at room temperature in this study, the stress temperature also highly affected the V_th_ shift by accelerating the activation of defects. Thus, the PBS- and NBS-induced instability could deteriorate as the stress temperature increases based on the stretched exponential function [42,43].

## 4. Conclusions

In conclusion, the LPA method is suggested for the low-temperature annealing of the InO_x_ semiconductor. The enhancement-mode amorphous InO_x_ TFTs were successfully fabricated using a solution process at a low process temperature of 200 °C. Through XPS analysis, LPA was able to greatly reduce the M–OH ratio, and the defective bonding states, in InO_x_ from 33% to 12.6%. As a result, the introduction of LPA leads to the significant improvement of μ_FE_ and SS from 0.13 to 0.81 cm^2^ V^−1^ s^−1^ and from 0.71 to 0.28 V dec^−1^, respectively. In addition, the LPA InO_x_ TFTs exhibited stable operation under PBS and NBS compared to the control device.

## Figures and Tables

**Figure 1 nanomaterials-13-02231-f001:**
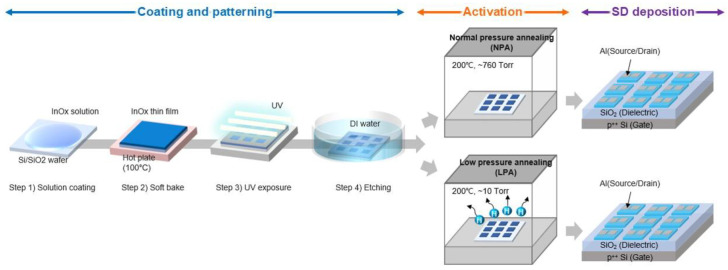
Fabrication process of low-temperature amorphous InO_x_ TFTs using LPA.

**Figure 2 nanomaterials-13-02231-f002:**
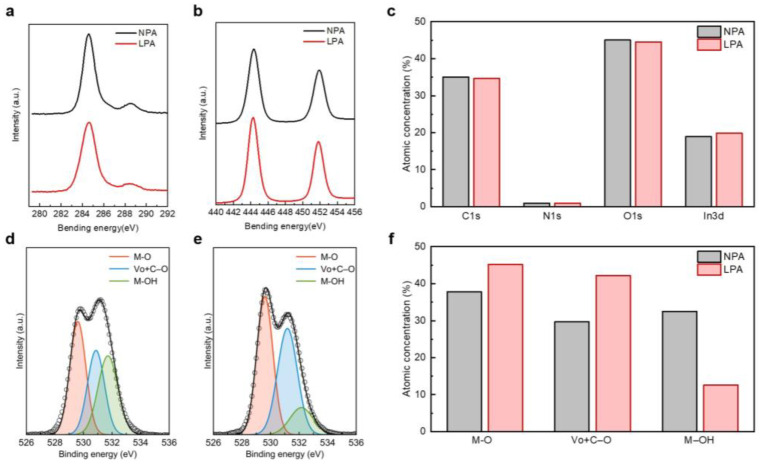
XPS (**a**) C 1s and (**b**) In 3d spectra of NPA and LPA InO_x_ thin films. (**c**) The comparison of the atomic percentages of C 1s, N 1s, O 1s, and In 3d of NPA and LPA InO_x_. XPS O 1 s spectra of (**d**) NPA InO_x_ and (**e**) LPA InO_x_ thin films. (**f**) The comparison of the atomic percentages of M–O, V_o_ + C–O, and M–OH of NPA and LPA InO_x_.

**Figure 3 nanomaterials-13-02231-f003:**
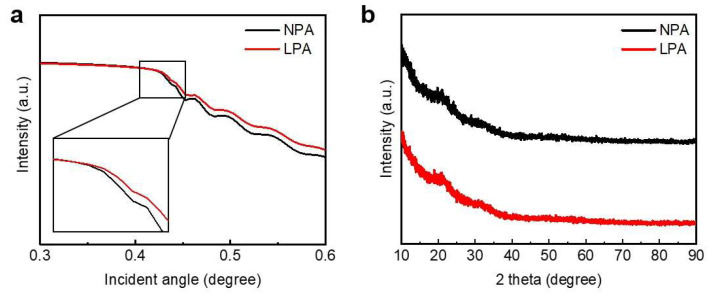
(**a**) XRR and (**b**) GIXRD spectra of NPA InO_x_ (black line) and LPA InO_x_ (red line).

**Figure 4 nanomaterials-13-02231-f004:**
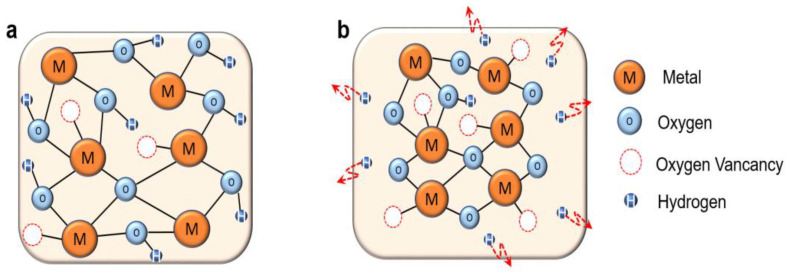
(**a**) NPA InOx TFT and (**b**) LPA InOx TFT, Proposed mechanism for the effect of LPA on the atomic structure and film formation of InO_x_.

**Figure 5 nanomaterials-13-02231-f005:**
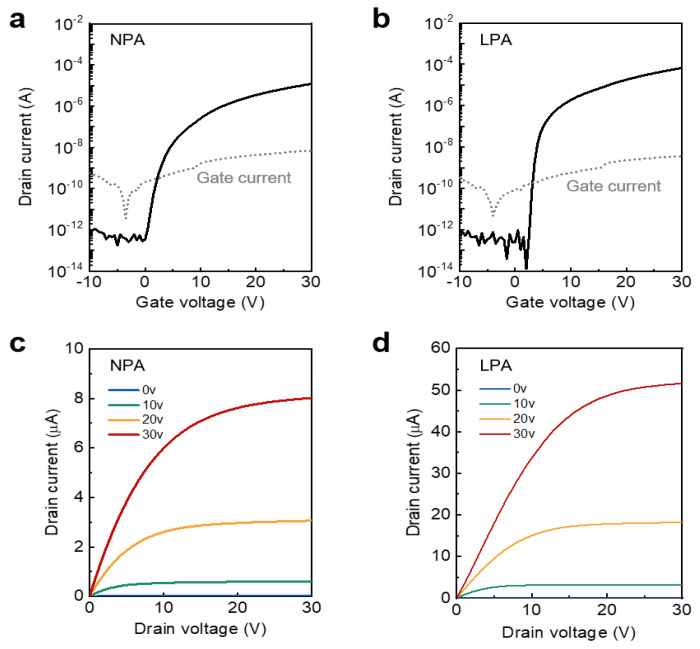
Transfer characteristics of (**a**) NPA InO_x_ TFTs and (**b**) LPA InO_x_ TFTs. The gray line indicates gate leakage current. Output characteristics of (**c**) NPA InO_x_ TFTs and (**d**) LPA InO_x_ TFTs.

**Figure 6 nanomaterials-13-02231-f006:**
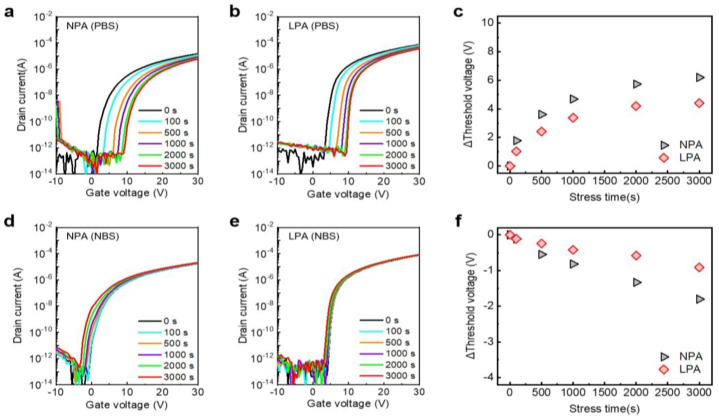
Transfer curve shifts of (**a**) NPA InO_x_ TFTs and (**b**) LPA InO_x_ TFTs under PBS. (**c**) Threshold voltage shift of NPA and LPA TFTs via PBS. Transfer curve shifts of (**d**) NPA InO_x_ TFTs and (**e**) LPA InO_x_ TFTs under NBS. (**f**) Threshold voltage shift of NPA and LPA TFTs via NBS.

**Table 1 nanomaterials-13-02231-t001:** Comparison with oxide TFTs.

Semiconductor	Dielectric	T (°C)	Vth (V)	μ_FE_ (cm^2^ V^−1^ s^−1^)	SS (V dec^−1^)	Ion/Off	Reference
InO_x_	SiO_2_	500	6.74	0.95	0.24	1.55 × 10^5^	[32]
InGaZnO	SiO_2_	450	−5.09	1.25	1.05	5.2 × 10^3^	[33]
ZnSnO	SiO_2_	350	10.7	0.58	-	~1 × 10^6^	[34]
ZnSnO	SiO_2_	600	-	1.12	0.39	3.0 × 10^7^	[35]
InO_x_	SiO_2_	240	-	1.57	0.45	3.9 × 10^7^	[36]
ZnSnO	SiO_2_	500	−0.95	2.41	1.25	3.9 × 10^6^	[37]
InSbO	SiO_2_	300	1.9	4.60	0.29	3 × 10^7^	[38]
InO_x_	SiO_2_	200	4.40	0.81	0.28	4.91 × 10^9^	This study

## Data Availability

Not applicable.

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
