# Peer review of "Low-Temperature Enhancement-Mode Amorphous Oxide Thin-Film Transistors in Solution Process Using a Low-Pressure Annealing"

_nanomaterials, 2023, doi:10.3390/nano13152231_

Round 1

Reviewer 1 Report

In this manuscript entitled “Low-Temperature Enhancement-Mode Amorphous Oxide Thin-Film Transistors in Solution Process Using a Low-Pressure Annealing”, Park et al. reported on a low-pressure annealing (LPA) method for preparing oxide thin-film transistors (TFTs). Impressively, the processing temperature is as low as 200 ℃. In addition, the LPA-derived InOx TFTs not only function in favorable enhancement mode but also demonstrate an outstanding on/off ratio exceeding 109. On the whole, the LPA strategy is cost-efficient, and the results are encouraging. This study will be of great significance for practical application. However, in the current stage, there is still room for further improvement. Therefore, a revision is recommended. The following comments should be fully taken into accounted.

1. In addition to the on/off switching ratio, subthreshold swing is also an important parameter for evaluating TFTs (e.g., Nanoscale, 2021,13, 5700-5705; Sci. Adv. 2022, 8, eabm9845). However, it is found that related parameter has been missed. It should be calculated.

2. As claimed in the introduction section by the authors, stability is a pivotal challenge of oxide TFTs. Therefore, I wonder if the TFTs produced by LPA have competitive stability or not in air after a long-term storage. It should be explored.

3. How is the field-effect mobility of the LPA derived TFTs? This is a very important performance metric. Please provide.

Reviewer 2 Report

The authors proposed the InOx TFTs by using low-pressure annealing (LPA) method. This method is effective and easy to operate. Therefore, this work has practical value. I have a few commets. 1:What is the stability of the device at high temperatures? 2:What is the comprehensive evaluation of the device performance compared with the reported device performance,? 3:Some abbreviations, such as NPA, are not fully spelled when they first appear.

Reviewer 4 Report

The manuscript is interesting. It proposes a simple method for improving the electrical properties of amorphous indium oxide films suitable for TFT fabrication.

77, 79 What determines the choice of the annealing time (2 h)? And if you make 24h NPA film, will it be better than 2h LPA?

98 The XPS peak corresponds to the release of an electron from the atom and cannot originate from a vacancy. Specifically, this peak corresponds to oxygen atoms near oxygen vacancies and in M–OC bonds (the latter is more probable). To clarify the situation, it is necessary to present the XPS data on carbon and indium atoms and/or the IR spectra of the films. It is unclear why, using the XPS method, the authors did not determine the exact composition of the films (how much In, O and C are they in)?

127 Give the values of Z, A and critical angles substituted into equation (1), explain how they were found. Estimate the error in the density values, taking into account the fact that the composition of the films is unknown, moreover, it is different for NPA and LPA InOx, the roughness of the films, which affects the value of the critical angle, was not controlled. Are the obtained density values 5.19 and 5.45 different or equal within the error?

153 Equation (2) contradicts the law of charge conservation! In addition, it is doubtful that the base (OH-) is formed by the action of an acid (H+), since OH- will immediately react with H+ to form water.

188 on Fig.5 c and d add gate voltage values

201,202 “…gate dielectric semiconductor interfaces…” should be replaced with “…gate dielectric InOx interfaces…” for clarity

205, 215 what is “oxygen peroxide”?

226, 227 the letters are confused in the signature to Fig. 6

It is necessary to give the values of the thicknesses of SiO2 and InOx.

It is necessary to specify the source of UV radiation and (at least approximately) estimate the power density or UV dose on the substrate.

The manuscript lacks a comparison of the results obtained by the authors with the literature data, for example, you can plot the subthreshold slope vs mobility (and / or on / off current ratios vs mobility), plot your points and points for similar oxides from the literature on it.

Round 2

Reviewer 1 Report

The authors have addressed all my previous concerns. Now it is suggested that this manuscript is ready for publication. 

Author Response

We appreciate to the reviewer's review.

Reviewer 2 Report

The author answered most of my questions. Another question, what are the possible ways to solve the problem of the temperature-derelated stability of oxide TFTs?

Reviewer 4 Report

106 “…metal–oxygen (M–O), oxygen vacancies (Vo), and metal hydroxide (M–OH)…” replaced by “…metal–oxygen (M–O), oxygen atoms near oxygen vacancies and in M– O–C [16] (Vo+C-O), and metal hydroxide (M–OH)…”.
163 Equation 2 is still wrong. An OH radical and an electron will immediately give an OH anion. I propose to remove it altogether and remove the text on lines 160 - 166.
166, 167 printed "...the high ratio of M-O weak bonds..." should be "...the high ratio of M-OH weak bonds...".
205 Add a description of table 1 to the text of the article.
244 extra dot.
